# The Effect of Age and Comorbidities: Children vs. Adults in Their Response to SARS-CoV-2 Infection

**DOI:** 10.3390/v16050801

**Published:** 2024-05-17

**Authors:** Girlande Mentor, Daniel S. Farrar, Costanza Di Chiara, Mi-Suk Kang Dufour, Silvie Valois, Suzanne Taillefer, Olivier Drouin, Christian Renaud, Fatima Kakkar

**Affiliations:** 1CHU Sainte-Justine, Département de Pédiatrie, Faculté de Médecine, Université de Montréal, Montreal, QC H3T 1C5, Canada; girlande.marie.carole.mentor@umontreal.ca (G.M.); o.drouin@umontreal.ca (O.D.); 2Centre for Global Child Health, The Hospital for Sick Children, Toronto, ON M5G 1E8, Canadacostanza.dichiara@sickkids.ca (C.D.C.); 3Division of Infectious Diseases, The Hospital for Sick Children, Toronto, ON M5G 1E8, Canada; 4Unité de Recherche Clinique Appliqué, Centre de Recherche du CHU Sainte-Justine, Montreal, QC H3T 1C5, Canada; mi-suk.kang.dufour.hsj@ssss.gouv.qc.ca; 5Centre D’infectiologie Mère-Enfant, Centre de Recherche du CHU Sainte-Justine, Montreal, QC H3T 1C5, Canada; s.v.hsj@ssss.gouv.qc.ca (S.V.);; 6Département de Microbiologie et Immunologie, Faculté de Médecine, Université de Montréal, Montreal, QC H3T 1C5, Canada; christian.renaud.hsj@ssss.gouv.qc.ca

**Keywords:** COVID-19, immunity, IgG antibody, pediatric infection

## Abstract

While children have experienced less severe coronavirus disease (COVID-19) after SARS-CoV-2 infection than adults, the cause of this remains unclear. The objective of this study was to describe the humoral immune response to COVID-19 in child vs. adult household contacts, and to identify predictors of the response over time. In this prospective cohort study, children with a positive SARS-CoV-2 polymerase chain reaction (PCR) test (index case) were recruited along with their adult household contacts. Serum IgG antibodies against SARS-CoV-2 S1/S2 spike proteins were compared between children and adults at 6 and 12 months after infection. A total of 91 participants (37 adults and 54 children) from 36 families were enrolled. Overall, 78 (85.7%) participants were seropositive for anti-S1/S2 IgG antibody at 6 months following infection; this was higher in children than in adults (92.6% vs. 75.7%) (*p* = 0.05). Significant predictors of a lack of SARS-CoV-2 seropositivity were age ≥ 25 vs. < 12 years (odds ratio [OR] = 0.23, *p* = 0.04), presence of comorbidities (vs. none, adjusted OR = 0.23, *p* = 0.03), and immunosuppression (vs. immunocompetent, adjusted OR = 0.17, *p* = 0.02).

## 1. Introduction

One of the most striking and least understood observations of the COVID-19 pandemic remains the difference in clinical manifestations and disease severity in children vs. adults [1]. Numerous cohort studies have described the increased severity of respiratory disease and the incidence of hospitalization and intensive care unit (ICU) admission in adults compared to children [2,3,4,5,6,7,8], which remained consistent throughout the emergence of SARS-CoV-2 variants of concern (VOC) [9,10]. The cause of these differences remains largely unknown. It is now clear that children are as easily infected as adults, as demonstrated by population-based seroprevalence data, in which children have shown a higher infection-induced seroprevalence than adults [11]. However, they remain largely asymptomatic or mildly symptomatic [12,13]. This is in marked contrast to other common respiratory infections such as influenza and respiratory syncytial virus (RSV), which typically affect younger children with greater severity [14].

While multiple hypotheses exist, it is suspected that differences in the immune response between adults and children play a role [15]. Children have had a more marked and dysregulated post-COVID-19 immune response, as seen by the novel multisystem inflammatory syndrome in children (MIS-C), primarily affecting children and, rarely, younger adults [16,17,18]. While data have emerged on both humoral and cell-mediated responses to infection in children versus adults, the results have been conflicting. Some studies have shown a more robust and durable humoral response to SARS-CoV-2 among children vs. adults [19,20], while others have shown a weaker and delayed response [21,22,23] and lower neutralizing activity [24]. Moreover, it is not clear which factors are predictive of this response, and whether they include host genetics and underlying comorbidities.

In this respect, studying the humoral response to infection within families infected by the same virus at the same time offers a unique opportunity to compare the immune response between children and adults, thereby adjusting for the possible confounding of results by time of infection, geography, variant, and potential genetic predisposition [25,26,27]. The objective of this study was therefore to compare the humoral immune response to SARS-CoV-2 infection in child vs. adult members of the same family, and to identify predictors of the magnitude of the response over time.

## 2. Materials and Methods

### 2.1. Study Design

This was a prospective cohort study (COVID-19 Family Cohort—COFAM) which enrolled children who tested positive for SARS-CoV-2 at the Centre Hospitalier Universitaire Sainte-Justine (CHUSJ) COVID-19 outpatient clinic in Montreal, Quebec, Canada from August 2020 to July 2021. From August to December 2020, the SARS-CoV-2 ancestral strain predominated in Quebec, with the first emergence of pre-Delta variants in January 2021. Variant identification was not possible during the study period. The COVID-19 clinic patient lists were reviewed weekly for positive cases, and families were contacted within 4 weeks of a positive test result. The CHUSJ is a tertiary-level maternal–child health center, and was the referral center for all children with COVID-19 in the Montreal area during this period. The study was approved by the CHUSJ Research Ethics Board. Written informed consent was obtained from each adult participant and by the parent or guardian of each child participant younger than 18 years of age.

### 2.2. Study Population Recruitment and Enrollment

Children aged < 18 years with polymerase chain reaction (PCR)-positive SARS-CoV-2 infection (i.e., index case) were offered enrollment alongside a symptomatic parent, sibling, or relative living in the same household who was also SARS-CoV-2 PCR-positive. After obtaining informed consent, a baseline questionnaire investigating sociodemographic characteristics, past medical history, and COVID-19-related symptoms and severity was administered via telephone interviews with the research nurse, and arrangements were made for blood sampling at home visits 6 and 12 months after infection in the index case. For all participants, a minimum of 3 mL of whole blood was drawn by venopuncture by the research nurse using an EDTA tube. These samples were transported at room temperature (within 4 h of sampling) to the CHU-SJ biobank, where samples were centrifuged to separate serum, and serum samples were stored frozen at -80 degrees Celsius and batched for analysis (see below).

### 2.3. SARS-CoV-2 Antibody Assays

Serum samples were tested on the DiaSorin LIAISON^®^ XL chemiluminescence analyzer using the DiaSorin Liaison SARS-CoV-2 S1/S2 IgG assay [28] in two separate batches (all 6-month samples and all 12-month samples were run together). This assay was the first SARS-CoV-2 serological test authorized by Health Canada and is a fully automated chemiluminescent immunoassay (CLIA) intended for the quantitative detection of IgG antibodies against SARS-CoV-2 spike protein receptor binding (S1) and fusion (S2) subunits in human serum and plasma. Anti-S1/S2 IgG antibody titers were expressed as arbitrary units per milliliter (AU/mL). Tests were performed according to the manufacturer’s protocol, namely that IgG concentrations < 12.0 AU/mL, between 12.0 AU/mL and 15.0 AU/mL, and ≥15 AU/mL up to 400 AU/mL were considered negative, equivocal, and positive, respectively. The assay’s reported sensitivity was 96.2% (94.2–97.7) and specificity was 98.9% (98.0–99.4), with a 94.4% agreement with plaque reduction neutralization tests (PRNT90) at a 1:40 ratio [29]. The global coefficient of variation was estimated to be between 2.9 and 5.3% according to the manufacturer and estimated to be 9% by the performing laboratory. Based on this assay, a minimum sample size of 70 (35 adults and 35 children) was selected to provide 80% power to detect an effect size of 0.2 in the antibody titer between groups, assuming an antibody threshold of at least 15/IU for all participants and using an alpha of 0.05.

### 2.4. Statistical Analysis

Participants were categorized as children (age < 18 years at the time of the positive SARS-CoV-2 PCR) and adults (age ≥ 18 years at the time of the positive SARS-CoV-2 PCR) for recruitment purposes. Given the potential variability in immune response within the pediatric age strata, and differences in the timing of vaccine approvals by age groups, participants were further sub-categorized as aged < 5 years (young children, no vaccine approved during the study period), aged 5–11 years (children, no vaccine approved during study period), aged 12–24 years (adolescents and young adults, vaccine approved during study period), and aged >25 years (older adults, vaccine approved during the study). Participants who were vaccinated prior to antibody testing were considered separately in the analysis.

Demographic and clinical characteristics are presented using descriptive statistics with reported frequencies and percentages, mean and standard deviation (SD), or median and interquartile range (IQR). Differences between children and adults were assessed using statistical tests that accounted for family clustering, including clustered χ^2^ tests and Somers’ D statistics. A statistical significance threshold of α = 0.05 was used for all analyses. COVID-19 severity was categorized as mild (no medical intervention or follow-up within 4 weeks of testing), moderate (medical day unit or hospitalization in pediatric ward for COVID-19 related diagnosis), and severe (hospitalization in intensive care for COVID-19-related diagnosis) using COVID-19 disease severity criteria from the World Health Organization and previously published information [30,31]. Median anti-S1/2 IgG titers were reported for all adults and children. Predictors of seropositivity were then assessed through unadjusted mixed-effects logistic regression with family ID included as a random effect and in a multivariate age-adjusted model. All analyses were performed using Stata, version 17.0.

## 3. Results

A total of 91 individuals (37 adults and 54 children) from 36 families were enrolled in this study. All participants completed the first serology testing (at 6 months post infection), while 82 participants (49 adults, 33 children) completed both 6- and 12-month testing. The first serology was performed at a mean time of 144 days (SD 50 days) after positive SARS-CoV-2 PCR, and the second serology was performed at a mean time of 359 days (SD 34 days) after positive SARS-CoV-2 PCR. None of the participants were vaccinated prior to the first serology; 13 participants (9 adults and 4 children) were vaccinated between the first and second serology assays.

### 3.1. Patient Demographics

The mean ages of all study participants at the time of the index cases’ positive PCR tests were 8.7 years [±5.8 SD] for children and 41.1 years [±11.4 SD] for adult participants. The demographic and clinical characteristics of the participants are detailed in Table 1. While there were no significant differences in sex or ethnicity between adult and pediatric participants, there was a larger proportion of adults vs. children with comorbidities (48.7% vs. 25.9%, *p* = 0.06), including hypertension (10.8% vs. 0, *p* = 0.03) and any immunodeficiency (21.6% vs. 5.6%, *p* = 0.02). Among symptomatic individuals, there was a significant difference in disease severity between adults and children; overall, adults had more severe disease (13.5% vs. 0%) or moderate disease (16.2% vs. 5.6%) (*p* = 0.01) (Table 1).

### 3.2. SARS-CoV-2 Antibody Response

Overall, only 78 (85.7%) participants were seropositive for anti-S1/S2 IgG antibody at six months following infection; the proportion of seropositive results was higher in children (n = 50, 92.6%) than in adults (n = 28, 75.7%) (*p* = 0.05). The median anti-S1/S2 IgG titer was higher in children vs. adults (107.0 AU/mL [IQR:51.8–164.0 AU/mL] in children vs. 79.8 AU/mL [IQR:18.8–219.0 AU/mL] in adults), although it was not statistically significant (*p* = 0.36). There was a significant difference in mean antibody titers across the extremes of ages, with a significant difference in mean antibody titers in participants < 5 years old [108.5 AU/mL (IQR: 20.4–188.0 AU/mL)] and 5–11 years of age [91.8 AU/mL (IQR: 48.2–128.5 AU/mL)] compared to participants >25 years of age [20.8 AU/mL (IQR: 7.6–107.5 AU/mL)], (*p* = 0.02 and *p* = 0.049, respectively). (Figure 1, Median antibody titers by age group among unvaccinated participants.)

### 3.3. Factors Predictive of Serological Response (Table 2)

The most significant predictors of absent seropositivity for anti-S1/S2 IgG were age ≥ 25 years (vs. <12 years; odds ratio [OR] = 0.23, 95% CI 0.06–0.94, *p* = 0.04), and after adjusting for age, the presence of comorbidities (vs. none; adjusted OR [aOR] = 0.23, 95% CI 0.06–0.87, *p* = 0.03), and presence of immunosuppression (vs. immunocompetent; aOR = 0.17, 95% CI 0.04–0.71, *p* = 0.02). Sex, ethnicity, and clinical disease severity did not have a significant effect on the antibody response to SARS-CoV-2 infection in either model. Specific details on participants who were seronegative at 6 months after infection are described in Appendix A. Among the 13 seronegative participants, 69% (9/13) had a chronic co-morbid condition, including 5 (38%) with a chronic viral infection.

**Table 2 viruses-16-00801-t002:** Predictors of SARS-CoV-2 seropositivity.

Characteristics	Crude Models			Age-Adjusted Models		
	OR	95% CI	*p* Value	aOR	95% CI	*p* Value
**Age category**						
<12 years	1 [Reference]	1 [Reference]	---	---	---	---
12–24 years	1.42	0.14–14.64	0.77	---	---	---
≥25 years	0.23	0.06–0.94	0.04	---	---	---
**Gender**						
Female	1 [Reference]	1 [Reference]	---	1 [Reference]	1 [Reference]	---
Male	1.26	0.36–4.47	0.72	1.24	0.34–4.52	0.74
**Ethnicity**						
White	1 [Reference]	1 [Reference]	---	1 [Reference]	1 [Reference]	---
Not White	1.81	0.53–6.18	0.34	2.04	0.57–7.28	0.27
**Comorbidity present**						
No	1 [Reference]	1 [Reference]	---	1 [Reference]	1 [Reference]	---
Yes	0.19	0.05–0.66	0.01	0.23	0.06–0.87	0.03
**Immunocompromised**						
No	1 [Reference]	1 [Reference]	---	1 [Reference]	1 [Reference]	---
Yes	0.13	0.03–0.54	0.005	0.17	0.04–0.71	0.02
**COVID-19 symptoms**						
Asymptomatic	1 [Reference]	1 [Reference]	---	1 [Reference]	1 [Reference]	---
Mild	1.27	0.30–5.42	0.74	1.59	0.35–7.15	0.55
Moderate/severe	1.20	0.17–8.38	0.85	3.26	0.35–30.53	0.30

### 3.4. Antibody Titers between 6 and 12 Months following Infection (Table 3, Figure 2)

The change in antibody response over time was assessed separately among vaccinated and unvaccinated participants. Among the 39 unvaccinated participants with serology at 6 and 12 months post infection, the majority (69%) had increasing antibody titers over time. The magnitude of the increase, however, was greater among adults than children. Among children < 5 years of age, 72.7% (8/11) increased their titers, with a median increase of 56.5 AU/mL (IQR 24.0–93.4) (Panel A); among children 5–11 years, only 60% (9/15) increased their titers (median increase of 41.7 AU/mL (IQR 16.3–51.0) (Panel B)). In comparison, 75% (3/4) of children 12–24 years increased their titers with a median increase of 299.0 AU/mL (IQR 252.0–317.5) (Panel C), and 77.8% (7/9) of adults ≥25 years of age increased their titers with a median increase of 338.6 AU (IQR 4.4–351.2) (Panel D). These differences in absolute titer increase across unvaccinated age groups was statistically significant (*p* = 0.006).

Among the 13 vaccinated participants with serology at 6 and 12 months post infection, all 4 of the participants between 5 and 24 years of age (100%) showed an increase in titers post vaccination (Panel E), while only 67% (6/9) adults ≥25 years showed increasing titers (Panel F). The magnitude of the increase in titers post vaccination was similar across age groups, with a median increase of 317.8 AU/mL (IQR 281–354.5) among children 5–11 years, compared to an increase of 347.2 AU/mL (IQR 322.7–371.7) among 12–24-year-old participants, vs. 306.6 AU/mL (IQR 181.0–372.3) among participants >25 years.

**Table 3 viruses-16-00801-t003:** Changes in antibody titers among unvaccinated vs. vaccinated participants with increasing titers.

Change in IgG Titer (AU/mL), Median (IQR)	Participants with Increased IgG, N	All Participants	Vaccination Status	
			Unvaccinated	Vaccinated
**Age category, PCR+ participants only**				
<5 years	8	+56.5 (24.0–93.4)	+56.5 (24.0–93.4)	---
5–11 years	11	+48.0 (16.3–58.0)	+41.7 (16.3–51.0)	+317.8 (281.0–354.5)
12–24 years	5	+317.5 (299.0–322.7)	+299.0 (252.0–317.5)	+347.2 (322.7–371.7)
≥25 years	13	+293.0 (153.6–351.2)	+285.0 (4.4–351.2)	+306.6 (181.0–372.3)

**Figure 2 viruses-16-00801-f002:**
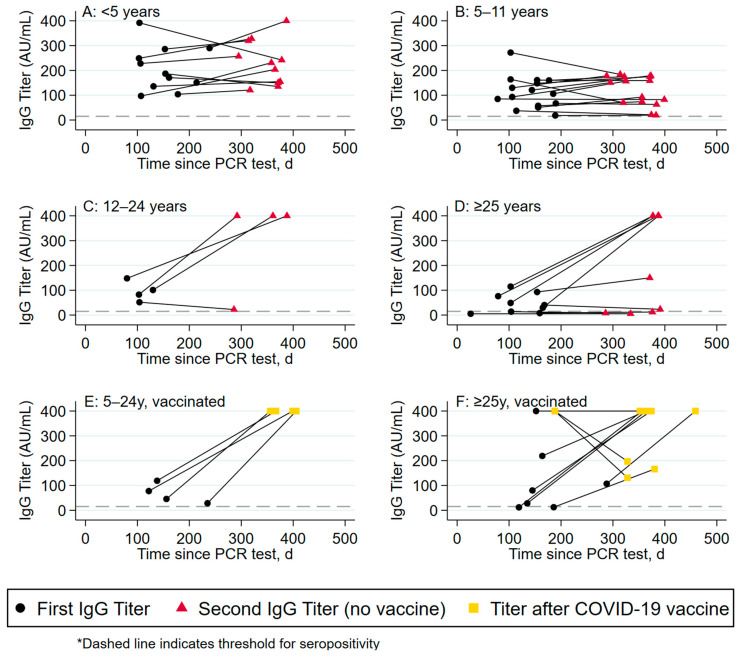
Change in antibody titers over time by age group.

## 4. Discussion

In this cohort of children and adults from the same family infected with SARS-CoV-2 during the first year of the pandemic, the most significant predictors of seroprotection following infection were age and the absences of comorbidities. Children had overall significantly less severe disease than adults, and yet had a more robust anti-SARS-CoV-2-specific humoral response with a higher proportion of seropositivity at 6 months following infection compared to adults, and a significant difference in the magnitude of response in very young children (age < 5) vs. older adults (age > 25). These data reinforce the differences in pediatric vs. adult manifestations of SARS-CoV-2, and highlight some of the novel elements in the immune response to infection in children vs. adults.

First, our data confirm previous findings that COVID-19 in children was significantly less severe than in adults [2,3,4,5,6,7,8]; while none of the children were categorized as severe and few (n = 3) as moderate, nearly 20% of the adults in this study had moderate or severe infection. Moreover, children had a more robust immune response to infection than adults, with a higher proportion seroprotected at 6 months following infection compared to adults (92.6 vs. 75.7%, *p* = 0.05). Taken together, these data highlight the uniqueness of SARS-CoV-2 infection compared to other viral infections such as influenza, in which increased antibody response is seen with increased severity of the disease [32,33], and RSV, in which children generally have lower circulating titers after infection compared to adults [34].

Consistent with previously published data on SARS-CoV-2, factors predictive of a lack of IgG response in this cohort included older age, the presence of comorbidities, and immunosuppression, which remained significant in the age-adjusted models [35,36,37]. Interestingly, the majority of participants with sequential 6- and 12-month tests showed an increasing antibody titer over time and durable protection at least 12 months after infection. This differs from previous reports in the adult population, which have shown the waning of post-infection antibodies and post-vaccination antibodies over 6–12 months [38]; however, this is consistent with newer pediatric data suggesting the durability of a response up to 18 months after infection [20,21]. We suspect that this may be due to our unique setting, with a shorter duration of school closures, such that children may have frequently been re-exposed to SARS-CoV-2 in school and/or childcare settings, resulting in a population with a higher incidence of potential natural boosting than the older, more isolated adults reported from studies in other settings. There was also a notable difference in the magnitude of the change in antibody response over time in children vs. adults. While children had higher total antibody titers at 6 months following infection, the increase in the titer over time was higher among unvaccinated adults than in children at 12 months following infection. While this may have been due to differences in the maturation of the adaptive immune response between children and adults [39], we cannot rule out some of the adults having been vaccinated without disclosure of their vaccination to the study team. Notably, there were participants within both group of adults, unvaccinated and vaccinated, that reached the upper limit threshold of the serological assay at 12 months, potentially suggesting vaccination as an explanation.

Our study had several limitations. First, the study occurred prior to the emergence of currently circulating variants and subvariants and therefore cannot be generalized to currently circulating SARS-CoV-2 variants. Second, our measures of immunity were limited to binding antibodies and were unable to determine the neutralizing abilities of these antibodies. Third, we did not assess baseline antibodies and the possibility of reinfection accounting for the differences between children and adults; however, the study occurred during the first wave of SARS-CoV-2, at which time it was unlikely that individuals had preexisting antibodies. Finally, reinfection between the 6- and 12-month serology assays may have been possible, although it was not captured, and may have contributed to increasing titers over time.

In summary, this study of COVID-19 within families highlights important differences in the durability and robustness of the humoral response to infection according to age and the presence of comorbidities. We demonstrated that children had significantly less severe disease, and yet produced significantly higher IgG responses initially, with older age and comorbidities the most significant predictors of a lack of seroprotection at 6 months following infection. While the mechanisms underlying this response are not well understood, these differences warrant exploration in larger cohort studies in which additional factors, such as the impact of sub-variants, may be accounted for. Nonetheless, given that these differences between children and adults remain to date a unique occurrence among common respiratory infections, these findings may help to further our understanding of protective immune mechanisms for other novel viral infections.

## Figures and Tables

**Figure 1 viruses-16-00801-f001:**
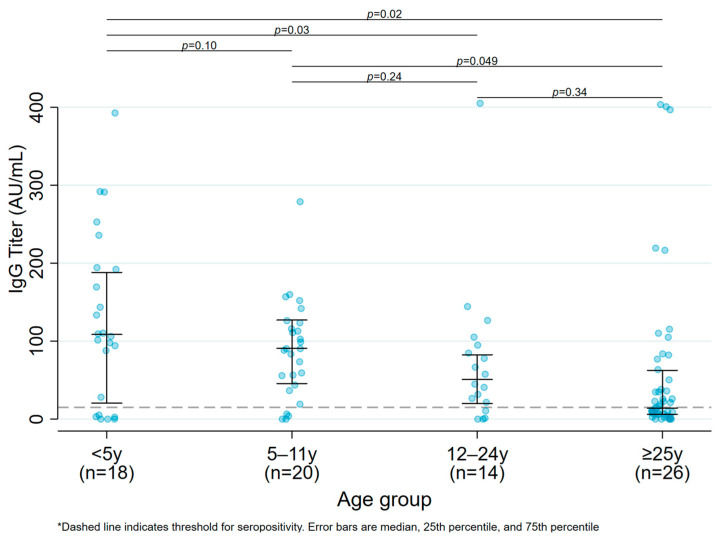
Median antibody titers by age group among unvaccinated participants.

**Table 1 viruses-16-00801-t001:** Demographic and clinical characteristics of adults and children.

Characteristics	Total All Families, n = 91		*p* Value
	Adults, (n = 37)	Children, (n = 54)	
**Age (years), mean (SD)**	41.1 (11.4)	8.7 (5.8)	NA
**Gender, n (%)**			1.00
Female	24 (64.9)	35 (64.8)	
Male	13 (35.1)	19 (35.2)	
**Ethnicity, n (%)**			0.89
Black	18 (48.7)	22 (40.7)	
White	7 (18.9)	18 (33.3)	
Arab/West Asian	7 (18.9)	8 (14.8)	
Other ^1^	5 (13.5)	6 (11.1)	
**Comorbidity present, n (%)**	18 (48.7)	14 (25.9)	0.06
Immunodeficiency	8 (21.6)	3 (5.6)	0.02
Autoimmune disease	4 (10.8)	1 (1.9)	0.12
Obesity	6 (16.2)	1 (1.9)	0.01
Hypertension	4 (10.8)	0 (0.0)	0.03
Asthma	3 (8.1)	2 (3.7)	0.51
Neurologic disease	0 (0.0)	3 (5.6)	0.22
Other	4 (10.8)	5 (9.3)	0.85
**SARS-CoV-2 symptoms**			0.31
Asymptomatic	5 (13.5)	13 (24.1)	
Symptomatic	32 (86.5)	41 (75.9)	
Mild	21 (56.8)	38 (70.4)	0.01
Moderate	6 (16.2)	3 (5.6)	
Severe/Critical	5 (13.5)	0 (0.0)	
**SARS-CoV-2 serology testing**			
Seropositive (Ab titer ≥15 AU/mL)	28 (75.7)	50 (92.6)	0.05
Titer (AU/mL), median (IQR)	79.8 (18.8–219.0)	107.0 (51.8–164.0)	0.36

^1^ Other includes Filipino and Latin American.

## Data Availability

Research data supporting this publication are available upon request.

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
