# Peer review of "The Effect of Age and Comorbidities: Children vs. Adults in Their Response to SARS-CoV-2 Infection"

_viruses, 2024, doi:10.3390/v16050801_

Round 1

Reviewer 1 Report

Comments and Suggestions for Authors

Comment and Suggestions for Author

The authors demonstrated that children had higher seropositivity rates compared to adults after SARS-CoV-2 infection. It is worth nothing that the severity of breakthrough infections and the seropositivity rates in subjects infected with SARS-CoV-2 is complex. SARS-CoV-2 seropositivity may vary widely among individuals based on multiple factors such as: socioeconomic and demographic risk factors, the specific SARS-CoV-2 variants/sub-variants, subject health status, preventive measures, cross-protection (or other coronavirus infection), the duration and strength of the immune response. Further research is needed with more homogenous groups, a larger sample size to evaluate variables which affect the rate of seropositivity against SARS-CoV-2 to understand the mechanisms underlying this phenomenon.

Below, please find other suggestions:                                                                                                                                                                The structure of the manuscript appears adequate and well divided in the sections.                                     

(i) ABSTRACT: (Please consider) While children have had less severe corona virus disease-2019 (COVID-19) after SARS-CoV-2 infection (COVID-19) than adults, the cause of this remains unclear. 

(ii) KEYWORDS: COVID-19; SARS-CoV-2 infection; SARS-CoV-2 immunity; IgG antibody; pediatric infection (The title words should not be repeated in Keywords)

(iii) Please include sufficient detail in Methods section so that experimental procedures can be repeated by other researchers in the field according to MDPI recommendations. I would recommend adding sample collection (whole blood sample via venipuncture – adult; for participants < 5 years of age - blood was collected by heel or finger stick?), processing, and storage. I would recommend adding the timing of serum collection relative to the onset of COVID-19 or an explanation. Please, explain if you were able to identify SARS-CoV-2 variants?

(iv) In “Statistical Analysis”: Participants were categorized as children (age <18 years at the time of the positive 97 SARS-CoV-2 PCR) and adults (age ≥18 years at the time of the positive SARS-CoV-2 PCR), 98 and further sub-categorized as age <5 years, 5-11 years, 12-24 years, and >35 years. In Figure 1, age group are /sub-categorized as age <5 years, 5-11 years, 12-24 years, and >35 years/, please explain it.

(v)The analysis has technical deficits: statistical analysis or cross-variable comparisons, and errors bars are lacking. 

(vi) Additional analysis would make this paper much stronger (SARS-CoV-2 variants/sub-variants between different families, re-infection, and COVID-19 vaccine during the study).

Comments on the Quality of English Language

English language is fine

Reviewer 2 Report

Comments and Suggestions for Authors

The effect of age and comorbidities: Children vs. adults in their 2 response to SARS-CoV-2 infection

The authors present a short but very well-written analysis of serological responses to COVID-19. I have very few comments.

(1) My main takeaway is that there was a statistically significant difference in the proportion of children seropositive (p = 0.05), but not a statistically significant difference in the titers (p = 0.36), but that the difference was more marked for the under 5s.

(2) I am not 100% sure what Figure 2 is telling me. Is the dashed line the threshold for seropositivity? If it is, then this should be described in the legend caption. Same would apply to Figure 1. The chart also seems to imply that the strongest Titers (after 12 months) are found in the subset of adults with an initial seropositive response, these people all seem to reach a Titer of ~400, stronger than children aged 5-11. This seems in opposition to the overall research conclusions? It is almost as if there are two "types" of adults in Figure 2, those that cannot generate a seropositive response at all (initially or after time) and those that generate an initially weak response that grows over time. Could the authors discuss this? Do lines 208-209 cover this point?

(3) As a very minor point, is it possible to change the shape as well as colour of first versus second IgG Titer in Figure 2? The legend is not so helpful when printed in B&W.

(4) Line 194, the authors probably mean "waning" rather than "weaning"

Overall, on reviewing this very short list of comments, it does seem like the manuscript might benefit from some fine tuning of its actual findings, for example whether there appear to be two "phenotypes" in the adult population of non / poor responders, and strong responders (and whether these correspond to reinfection). Also whether the cut-off is not so much children, as under-5s again having a different "phenotype" to the rest. It may also be worth considering whether there is any risk that some of the 39 people not receiving vaccinations, might actually have been vaccinated (but not disclosed it), as this would be a possible explanation for why adults have such a strong Titer at 12 months. Out of curiosity, what was the Titer for adults who did receive a vaccination, at 12 months? If it was also 400 AU/mL then this would be suggestive of some problems with vaccination disclosure.

Round 2

Reviewer 1 Report

Comments and Suggestions for Authors

I would like to thank the Authors for addressing my initial comments. I clearly support the publication of this manuscript in a revised form in this journal.

Reviewer 2 Report

Comments and Suggestions for Authors

The authors have comprehensively responded to my comments, and I appreciate their efforts very much, I believe the manuscript should be published.